



# Response of the carbon cycle to the different orbital configurations of the last 9 interglacials

Nathaelle Bouttes[1], Didier Swingedouw[1], Didier M. Roche[2,3], Maria F. Sanchez-Goni[1,4], Xavier Crosta[1]

[1] Univ. Bordeaux, EPOC, UMR 5805, F-33615 Pessac, France
[2] Laboratoire des Sciences du Climat et de l'Environnement, LSCE/IPSL, CEA-CNRS-UVSQ, Université Paris-Saclay, F-91191 Gif-sur-Yvette, France
[3] Earth and Climate Cluster, Faculty of Earth and Life Sciences, Vrije Universiteit Amsterdam, Amsterdam, The Netherlands
[4] EPHE, PSL Research University, F-33615 Pessac, France

*Correspondence to*: Nathaelle Bouttes (nathaelle.bouttes@lsce.ipsl.fr)

**Abstract.** Atmospheric $CO_2$ levels during interglacials prior to the Mid Bruhnes Event (MBE, ~430 ka BP) have lower values
of around 40 ppm than after the MBE. The reasons for this difference remain unclear. A recent hypothesis proposed that
changes in oceanic circulation, in response to differences in external forcing before and after the MBE, might have increased
the ocean carbon storage and thus explained the lower $CO_2$. Nevertheless, no quantitative estimate of this hypothesis has been
produced up to now. Here we use an intermediate complexity model including the carbon cycle to evaluate the response of the
carbon reservoirs in the atmosphere, ocean and land in response to the changes of orbital forcings and atmospheric $CO_2$
concentrations over the nine last interglacials. We show that the ocean takes up more carbon during pre-MBE interglacials in
agreement with data, but the impact on atmospheric $CO_2$ is limited to a few ppm. Terrestrial biosphere is simulated to be less
developed in pre-MBE interglacials, which reduces the storage of carbon on land and increases atmospheric $CO_2$. Accounting
for different simulated ice sheet extents modifies the vegetation cover and temperature, and thus the carbon reservoir
distribution. Overall, atmospheric $CO_2$ is slightly smaller in these pre-MBE simulated interglacials including ice sheet
variations, but the magnitude is still far too small. These results suggest a possible mis-representation of some key processes
in the model, such as the magnitude of ocean circulation changes, or the lack of crucial mechanisms or internal feedbacks,
such as those related to permafrost, that could explain the lower atmospheric $CO_2$ concentrations during pre-MBE interglacials.

## 1 Introduction

Ice core data have shown that atmospheric $CO_2$ concentration during interglacials of the last 800,000 years has changed
(Luthi et al., 2008; Bereiter et al., 2015). Older interglacials before the Mid-Bruhnes Event (MBE) around 430 ka BP, i.e.





Marine Isotope Stage (MIS) 13, 15, 17 and 19, are characterised by relatively lower atmospheric $CO_2$, around 240 ppm, compared to more recent interglacials, i.e. MIS 1, 5, 7, 9 and 11, which have a higher $CO_2$ level of around 280 ppm (Figure 1a).

Proxy data such as the marine $\delta^{18}O$ stack record, embedding both deep-sea temperature and ice-sheet volume (Lisiecki and
Raymo, 2005), indicate that older interglacials (pre-MBE) have a colder climate than the more recent ones (post-MBE). This tendency is also supported by individual $\delta^{18}O$ and sea surface temperature (SST, derived from Mg/Ca paleothermometry, alkenones or foraminifera assemblages) records from marine sediment cores (Lang and Wolff, 2011; Past Interglacials Working Group of PAGES, 2016), although some individual sub-stages such as MIS 7c and 7e do not follow this general tendency.

Numerical simulations with an intermediate complexity model have demonstrated that differences in Earth's orbital configuration, and hence seasonal and spatial distribution of insolation, cannot explain alone the colder climate recorded during pre-MBE interglacials, whereby lower atmospheric $CO_2$ concentration is also necessary to simulate colder climate (Yin and Berger, 2010; 2012). However, the reasons for the lower $CO_2$ values remain elusive. In that respect, transient simulations of the last 740,000 years with a box model have shown that the lower interglacial $CO_2$ values prior to the MBE
might be explained by lower ocean temperature and weaker thermohaline circulation compared to post-MBE interglacials (Köhler et al., 2006). Another hypothesis proposed that vigorous bottom water formation and stronger ventilation in the Southern Ocean during pre-MBE interglacials could increase deep oceanic carbon storage and lower atmospheric $CO_2$ (Yin, 2013). However, this has not been evaluated yet in a climate model including a carbon cycle representation. In addition, changes in surface temperature also modify the partition of the carbon cycle: in the ocean, colder sea surface temperature
increases the solubility of $CO_2$, increasing its potential uptake from the atmosphere during pre-MBE interglacials. In contrast, on land a colder climate might yield a decrease in biomass reducing $CO_2$ uptake *via* lower continental carbon storage. Because the ice sheets in the North Hemisphere are different during the interglacials in response to the different values of $CO_2$ and orbital configurations (Ganopolski and Calov, 2011), they might also have an impact on the carbon cycle, for example by modifying the terrestrial biosphere extent.

Here, we test the impact of the different orbital configurations of the last nine interglacials on the carbon cycle. For this purpose, we use a coupled carbon-climate model to evaluate the changes of carbon storage in the ocean and in the terrestrial biosphere, as well as the impact of different North Hemisphere ice sheet volumes.

## 2 Methods

We use the iLOVECLIM climate model of intermediate complexity, which is a new development branch (code fork) of the LOVECLIM model in its version 1.2 (Goosse et al., 2010). iLOVECLIM has an atmosphere module (ECBILT) with a T21 spectral grid truncation (~5.6° in latitude/longitude in the physical space) and 3 vertical layers. The ocean component (CLIO)

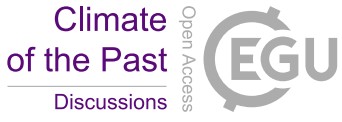

has a horizontal resolution of 3° by 3° and 20 vertical levels. The evolution of the terrestrial biosphere, i.e. the proportion of desert, grasses and tree cover, is computed by the VECODE model (Brovkin et al., 1997). It includes a carbon cycle module on land and in the ocean (Bouttes et al., 2015). iLOVECLIM is an evolution from the LOVECLIM model used in previous model studies of the last nine interglacials focused on climate (Yin and Berger, 2010; 2012; Yin, 2013). It has the same

atmospheric and oceanic modules, but includes a different carbon cycle representation in the ocean (Bouttes et al., 2015). We have chosen the same dates for the nine orbital configurations as in Yin and Berger (2010; 2012) and Yin (2013), *i.e.* the maximum of insolation preceding the $\delta^{18}O$ peak values (Table 1, Figure 1b and c). Contrary to most simulations from these studies, we also use the $CO_2$ values (as well as $CH_4$ and $N_2O$) at the same dates as for the orbital configurations (and not at the $CO_2$ peak), but as stated in Yin and Berger (2012), this may not affect the main results concerning the simulated climatic

changes.

| MIS | Date of $\delta^{18}O$ peak (ka BP) | Date for orbital configuration and $CO_2$ (ka BP) | $CO_2$ values from data (ppm) |
|---|---|---|---|
| 1 | 6 | 12 | 243.2 |
| 5.5 | 123 | 127 | 268.64 |
| 7.5 | 239 | 242 | 269.23 |
| 9.3 | 329 | 334 | 280.32 |
| 11.3 | 405 | 409 | 282.29 |
| 13.13 | 501 | 506 | 235.92 |
| 15.1 | 575 | 579 | 249.36 |
| 17 | 696 | 693 | 234.38 |
| 19 | 780 | 788 | 242.73 |

Table 1 Dates of orbital parameters and $CO_2$ used for the simulations (Luthi et al., 2008).

In the model, we separate the atmospheric $CO_2$ concentration into two distinct variables depending on its physical and chemical impact. The first one is used in the radiative scheme of the atmosphere, for which we prescribe in all the described simulations the $CO_2$ from measured values (Lüthi et al., 2008; Figure 1a). Another atmospheric $CO_2$ is computed

interactively in the model, as a result of the balance of the carbon fluxes between the different carbon sub-components (atmosphere, ocean and terrestrial biosphere). We make this choice of keeping the two separated to ensure that the climate simulated by the model is coherent with past measured atmospheric $CO_2$. In other words, we consider the atmospheric $CO_2$ concentration as an imposed external forcing, while within the carbon cycle the atmospheric $CO_2$ concentration is allowed to vary, but does not impact the atmospheric radiative forcing. By doing this, we limit the number of degrees of freedom in our

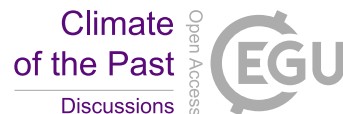

climate carbon system, which notably allows to avoid the complication arising from simulating a different climate when the climate-carbon is fully coupled.

The simulations performed are snapshots, run with constant orbital and atmospheric $CO_2$ concentration forcing and integrated over 3000 years allowing the ocean to reach a quasi-equilibrium. All simulations start from the pre-industrial

control one, and the average of the last 100 years is used to analyse the results.

Our strategy is to evaluate the impact of the different climate and carbon compartments to set the atmospheric $CO_2$ concentration. For this purpose, we consider three series of simulations, which have all been run for the nine interglacials (Table 2). The first series (OC "Ocean Carbon") has fixed ice sheets set to the observed pre-industrial ones and fixed terrestrial biosphere set to the simulated pre-industrial one. This first set of simulations thus provides the response of the

ocean alone to the different orbital parameters and $CO_2$ levels of the nine interglacials. The second series (OVC "Ocean Vegetation Carbon") has still fixed ice sheets, but includes an interactive terrestrial biosphere, computed by the model. It gives the response of both the ocean and land vegetation reservoirs to the different orbital parameters and $CO_2$ as well as their interactions for setting the atmospheric $CO_2$ concentration. Finally, the third series (OVIC "Ocean Vegetation Ice sheet Carbon") has different prescribed ice sheets in the North Hemisphere for the nine interglacials. The ice sheet distribution

change is based on modelling results, given that the uncertainty from data is very large for the interglacials of the last 800,000 years, especially the oldest ones. The ice sheet distributions are thus taken from an ice sheet simulation of the last 800,000 years with the CLIMBER-2 model and its ice sheet component SICOPOLIS (Ganopolski and Calov, 2011). The ice sheet distribution is chosen 2,000 years after the chosen interglacial date to account for the long timescale of the ice sheet response during a deglaciation and ensure that the ice sheet corresponds to an interglacial configuration. The ice sheet

elevations for the nine interglacial simulations are shown on Figure 2. The terrestrial biosphere is also interactive in this OVIC series of simulations. This last set of simulations thus adds the effect of having different ice sheets in the North Hemisphere for the carbon cycle variations.

| Name of the series | Components impacting the carbon cycle | | |
| --- | --- | --- | --- |
| | Ocean | Vegetation | Different interglacial ice sheets |
| OC | ✓ | ✗ | ✗ |
| OVC | ✓ | ✓ | ✗ |
| OVIC | ✓ | ✓ | ✓ |





Table 2. Summary of the three series of simulations.

## 3 Results and discussion

### 3.1 Role of the ocean (OC simulations)

Similar to previous numerical studies of the interglacials with the LOVECLIM model (Yin and Berger, 2010; 2012), the changes in orbital configuration and atmospheric $CO_2$ lead to altered sea surface temperature and oceanic circulation for each interglacial simulation of the OC series. All simulations are warmer than the control pre-industrial in the high latitudes in the North Hemisphere (Figure 3). Except for MIS1, the post-MBE simulations (corresponding to MIS 5, 7, 9 and 11) are also warmer than the pre-industrial control in large areas in the mid latitudes of the North Hemisphere and MIS 5, 9 and 11 are

slightly warmer in the Southern Ocean. In the pre-MBE simulations (MIS 13, 15, 17 and 19), the ocean is mainly colder than the pre-industrial, especially in the South Hemisphere. To compare the pre-MBE to post-MBE simulations, we built a composite (average) for each period (pre- and post-MBE). We have excluded MIS 1 from the post-MBE composite, for which the date chosen corresponds to a $CO_2$ much lower than the other post-MBE interglacials. We thus consider MIS 5, 7 9 and 11 in the post-MBE composite and MIS 13, 15, 17 and 19 in the pre-MBE composite. The difference between the pre-

and post-MBE composites show that the pre-MBE interglacial simulations are colder in the surface ocean than the post-MBE ones, especially in the Southern Ocean (Figure 4a). This is in general agreement with SST data, which indicate colder SST in the pre-MBE interglacial oceans, especially in the Southern Ocean (Table 3 and figure 4a).

| latitude | longitude | site | MIS 5e | MIS 7e | MIS 9e | MIS 11c | MIS 13a | MIS 15a | MIS 17c | MIS 19c | post-MBE | pre-MBE | Difference pre-post MBE |
|---|---|---|---|---|---|---|---|---|---|---|---|---|---|
| 57.51 | -15.85 | ODP 982 | 16.2 | 14.5 | 15.8 | 15 | 13.7 | 14.1 | 14.2 | 14.1 | 15.375 | 14.025 | -1.35 |
| 56.04 | -23.23 | DSDP 552s | 15.1 | 14.7 | 14.2 | 16.4 | 12.4 | 14.7 | 18.3 | 14.7 | 15.1 | 15.025 | -0.075 |
| 41.01 | -126.43 | ODP 1020 | 14.1 | 11.7 | 12.8 | 14 | 10.2 | 12.5 | 13.6 | 12.1 | 13.15 | 12.1 | -1.05 |
| 41.00 | -32.96 | DSDP 607s | 25.1 | 20.5 | 23.6 | 26.8 | 22.3 | 20.3 | 25.2 | 24 | 24 | 22.95 | -1.05 |
| 32.28 | -1148.40 | ODP 1012 | 19.5 | 17.7 | 19.7 | 19.1 | 17.5 | 18.3 | 19.3 | 18 | 19 | 18.275 | -0.725 |




| 19.46 | 116.27 | ODP 1146 | 27.3 | 26.3 | 27.3 | 26.8 | 26.1 | 26.3 | 26.9 | 26.2 | 26.925 | 26.375 | -0.55 |
|---|---|---|---|---|---|---|---|---|---|---|---|---|---|
| 16.62 | 59.80 | ODP 722 | 27.7 | 27.3 | 27.5 | 27.5 | 27 | 27.1 | 27.2 | 27.2 | 27.5 | 27.125 | -0.375 |
| 9.36 | 113.29 | ODP 1143 | 28.8 | 27.8 | 28.6 | 28.3 | 28.4 | 28.1 | 28.6 | 28.2 | 28.375 | 28.325 | -0.05 |
| 2.04 | 141.76 | MD97-2140 | 29.5 | 28.6 | 29 | 29.5 | 28.6 | 28.4 | 29.3 | 28.9 | 29.15 | 28.8 | -0.35 |
| 0.32 | 159.36 | ODP 806B | 29.6 | 29.2 | 28.8 | 30.2 | 28.2 | 29.4 | 29 | 29.4 | 29.45 | 29 | -0.45 |
| -3.10 | -90.82 | ODP 846 | 25.1 | 24 | 23.8 | 24 | 23.6 | 23.7 | 23.7 | 23.7 | 24.225 | 23.675 | -0.55 |
| -41.79 | -171.50 | ODP 1123 | 17.7 | 19 | 19.6 | 19.3 | 17.8 | 18.8 | 18 | 17.9 | 18.9 | 18.125 | -0.775 |
| -42.91 | 8.9 | ODP 1090 | 17.1 | 10.2 | 14.7 | 13.9 | 10.2 | 11.7 | 11.1 | 10.4 | 13.975 | 10.85 | -3.125 |
| -43.45 | 167.9 | MD06-2986 | 18 | 16.5 | 16.6 | 18.1 | 15.5 | 16.2 | 16.3 | 15.8 | 17.3 | 15.95 | -1.35 |
| -45.52 | 174.95 | DSDP594 | 18.3 | 7.1 | 9.5 | 17.5 | 10 | 11.7 | 12.1 | 9.7 | 13.1 | 10.875 | -2.225 |

Table 3. SST data from Past Interglacials Working Group of PAGES (2016) and shown on Figures 4a, 13a and 16.

Compared to the preindustrial, the ventilation of the Southern Ocean is increased in all simulations (Figure 5). The formation of AABW as well as the wind driven meridional cell between 40 and 60°S (so called Deacon cell) are both stronger. On

average, the maximum of the Deacon cell is increased by 7% between pre and post MBE simulations, while AABW is increased by 18% (Figure 4b). The meridional overturning circulation is also slightly increased by 6% and deepened in the Atlantic Ocean. All these results concerning oceanic circulation changes are very similar to those of Yin et al. (2013), allowing to test their hypothesis on the impact of these changes on ocean carbon uptake and atmospheric $CO_2$ concentrations. The change of circulation and sea surface temperature modifies the uptake of carbon into the ocean. The colder sea surface

temperature of the pre-MBE simulations, which increases dilution of $CO_2$ at the ocean surface, associated with stronger ventilation, yields a larger carbon uptake by the ocean. This results in higher dissolved inorganic carbon (DIC) concentrations in the Southern Ocean, as well as higher DIC concentration in the upper ocean (first 2 km of the ocean) (Figure 4c), reflecting the global increase of carbon storage in the ocean of 4.7 GtC on average for pre-MBE simulations. On the opposite, the DIC slightly decreases in the deeper ocean, which may be due to the increased ventilation of NADW

bringing more carbon back from the deep ocean to the surface.





The stronger uptake of carbon by the ocean in the old interglacials leads to a decrease in $CO_2$ and lower values for the pre-MBE interglacials compared to the post-MBE ones (Figure 6), in agreement with $CO_2$ data, as shown by the very good correlation between the measured and simulated values (r=0.91, p<0.01, Fig. 6). However, the difference in magnitude between pre and post MBE values is very small in the simulations and accounts only for a few ppm. Thus, even though it

goes qualitatively in the same direction as the data, with lower values for the older interglacials, the magnitude of the difference is much larger in the data with a ~30-40 ppm difference while the maximum difference is ~1-5 ppm in the simulations. In fact, the slope of the linear regression between simulated atmospheric $CO_2$ concentration and observed ones is of 0.07, indicative of an underestimation of more than 14 times in the simulations.

Hence the ocean carbon uptake in the simulations is not sufficient to drive a significant lowering of $CO_2$ as seen in the

measured data. Either the change of circulation and surface temperature should be larger, or another mechanism and feedbacks need to be taken into account to modify the biological or physical carbon uptake and amplify the initial change. Since the representation of bottom water formation in the Southern Ocean is biased in the model with an overrepresentation of open ocean convection, as is also the case for many more complex General Circulation Models (GCMs) (Heuzé et al., 2013), it is possible that this hinders simulating the full range of carbon storage due to circulation changes, as it is suspected

for colder periods such as the Last Glacial Maximum (around 21,000 years ago) (Fischer et al., 2010). In particular, for the colder pre-MBE interglacials as for the LGM, it is possible that modifying the simulation of sinking of bottom water in the Southern Ocean due to brine release during the formation of sea ice would result in more carbon stored in the ocean (Bouttes et al., 2010).

### 3.2 Role of land vegetation and soils (OVC simulations)

In the first series of simulations, only the ocean was allowed to respond to the different external forcings, while the vegetation and soils on land were fixed to the pre-industrial distribution. To account for changes in land vegetation and soils on the carbon cycle, the second series of simulations (OVC) has an interactive terrestrial biosphere module on top of the ocean one (Table 2).

Compared to the pre-industrial control simulation (and the OC series with fixed terrestrial biosphere), more trees develop in North Africa and the southern part of Eurasia in these interglacial simulations, while the tree cover is reduced in central North America and some regions in the northern part of Eurasia (Figure 7). When compared to the simulations with fixed vegetation, the interactive vegetation leads to ocean surface warming almost everywhere except in the North Atlantic for some interglacials (Figure 8). In response to this general warming of the surface ocean, the stratification in the convection

region is increased (*e.g.* Swingedouw et al., 2007) leading to a slowdown of the Atlantic Meridional Oceanic Circulation (AMOC), especially for MIS 1, 5 7, 9 and 15 (Figure 9).

For the carbon cycle, the activation of the terrestrial module results in more carbon stored in the vegetation and soils for all interglacial simulations (Figure 10b) since the vegetation cover increases compared to the control because of the warmer

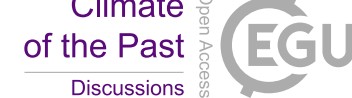



climate. This tends to lower atmospheric $CO_2$ concentration, hence the $pCO_2$ difference at the air-sea interface, leading to an outgassing of carbon from the ocean to the atmosphere, which ultimately decreases the storage of carbon in the ocean. The ocean carbon storage is also diminished compared to the series of simulations with fixed vegetation due to the warmer ocean temperature, which reduces the $CO_2$ solubility in water. The increase of carbon storage in the terrestrial biosphere is

generally larger than the loss of carbon from the ocean so that the carbon content of the atmosphere is also diminished in these simulations compared to the fixed vegetation simulations, and atmospheric $CO_2$ is slightly lower or not changed (Figure 11a).

In terms of difference between pre and post MBE interglacial simulations, we find less vegetation cover in most areas for the older interglacials that are colder (except in North Africa and parts of south Eurasia) and consequently less carbon stored (-

48 GtC) in the vegetation and soils in the pre-MBE simulations compared to post-MBE simulations (Figure 12). This effect tends to increase atmospheric $CO_2$ on average in pre-MBE interglacial simulations.

For the ocean, the differences between pre- and post-MBE simulations are similar to the ones for the simulations with fixed vegetation (OC). On average, the sea surface temperature is lower in the pre-MBE simulations compared to post-MBE simulations except in a small area in the North Atlantic (Figure 13a) and the ventilation is increased in the pre-MBE

simulations (Figure 13b). Hence the ocean can store more carbon in the pre-MBE simulations than the post-MBE simulations with an average increase of carbon storage in the pre-MBE ocean of 43 GtC compared to the post-MBE ocean. Similarly, the DIC concentration is higher in pre-MBE simulations, especially in the upper ocean and deep Southern Ocean, as in the previous series of simulations with fixed vegetation (Figure 13c).

As the diminution of carbon storage by the terrestrial biosphere in the pre-MBE simulations is larger than the increase of

carbon storage by the ocean, it results in more carbon in the atmosphere and higher $CO_2$ on average for the pre-MBE simulations than the post-MBE simulations (Figure 11b), which is thus in qualitative disagreement (negative correlation of -0.33 (p=0.38) between simulated and observed atmospheric $CO_2$ for the interglacials considered) with the observations. Nevertheless, it should be noted that permafrost (frozen soil) was not taken into account in these simulations. If there was more permafrost during the colder pre-MBE interglacials, it could store more carbon on land and counteract the loss of

carbon due to the lowering of vegetation cover and production (Crichton et al., 2016).

Comparison with pollen data (Table 4) indicates that the model is in qualitative agreement with reconstructed tree cover change in South America where the tree cover was smaller on average in pre-MBE than in post-MBE interglacials (Figure 12a). In southern Europe the tree fraction data also indicate that less tree cover prevailed during pre-MBE than during post-MBE interglacials, this time contrasting with the simulations which are not correctly representing the change of tree cover.





| | | SW Iberian margin (MD95-2042, MD01-2443, IODP U1385) Tree cover = Mediterranean Forest pollen % | | Tenaghi-Phillipon Tree cover = Temperate Forest pollen % | | Funza Arboreal pollen%-Quercus % | |
|---|---|---|---|---|---|---|---|
| | | Interglacial values | Average | Interaglacial values | average | Interglacial values | average |
| Post-MBE | MIS5e | 68 | | 96 | | 86 | |
| | MIS7e | 42 | | 92.4 | 95.9 | 75 | 76.7 |
| | MIS9c | 54 | | 95.5 | | 73 | |
| | MIS11c | 48 | 53.0 | 99.5 | | 73 | |
| Pre-MBE | MIS13a | 48 | | 95.8 | | 73 | |
| | MIS15a | 48 | 44.5 | 96.9 | 89.5 | 65 | 71.2 |
| | MIS17c | 35 | | 82.2 | | 78 | |
| | MIS19c | 47 | | 83.3 | | 69 | |
| Difference Pre-MBE – post-MBE | | | -8.5 | | -6.4 | | -5.5 |

Table 4: Tree cover (%) reconstructed from pollen data in three sites. SW Iberian margin: MIS 5 (MD95-2042, Sanchez Goñi et al., 1999), MIS 7 (MD01-2443, Roucoux et al. 2006), MIS 9, 13, 15, 17 (IODP 1385, unpublished data), MIS11 (U1385, Oliveira, 2016), MIS 19 (U1385, Sanchez Goñi et al., 2016); Tenaghi-Phillipon, Greece (Past Interglacials Working Group of PAGES, 2016); and Funza, Colombia (Past Interglacials Working Group of PAGES, 2016).

### 3.3 Impact of different ice sheets (OVIC simulations)

The last series of simulations (OVIC) has the same design as OVC but also takes into account possible differences in ice sheet distribution in the North Hemisphere based on numerical simulations (Ganopolski and Calov, 2011). The simulated ice sheet distributions used for our interglacial simulations mainly differ in North America. On average, the North American ice

10  sheet is more extended in the pre-MBE interglacials compared to the post-MBE interglacials (Figure 14).

The change of ice sheet extent has a large regional impact on vegetation cover, which is reduced where the ice sheet extends more. On average, it results in a reduction of vegetation in North America in the pre-MBE interglacials, when the ice sheet is more extended, compared to the post-MBE interglacials (Figure 15).



The increase of ice sheet extent and diminution of vegetation cover for pre-MBE simulations has two main impacts for the carbon cycle: (i) it diminishes the terrestrial biosphere carbon storage, increasing atmospheric $CO_2$, but (ii) it also cools the climate due to the higher albedo from the ice. Consequently, the ocean temperature decreases, especially in the North Hemisphere (Figure 16 compared to Figure 13a) and the ocean carbon storage increases, which lowers atmospheric $CO_2$.

This second effect dominates and the overall result is a lower atmospheric $CO_2$ in pre-MBE simulations compared to post-MBE simulations (Figure 11). As for the other processes analysed, it only modifies atmospheric $CO_2$ by a few ppm, though correcting back (compared to OVC simulations) the difference pre-MBE minus post-MBE towards the observations. Nevertheless, the correlation between simulated and measured $CO_2$ (accounting for MIS1) is very small (-0.08) and not significant (p=0.83). The magnitude of the changes of atmospheric $CO_2$ among the different interglacials is once again

largely underestimated as compared to observations.

    Accounting for different ice sheets in the OVIC series seems to improve the model-data comparison in southern Europe for tree cover (Figure 15a) where the data are at the limit between regions of more tree coverage and less tree coverage in the model. This highlights the role of ice sheet extent in setting the vegetation pattern. Nevertheless, the uncertainty in ice sheet distribution is very large and the model-based reconstruction might not be accurate. For example, the lack of IRD (Ice Rafted

Debris) from North America before MIS16 and the presence of IRD from Europe indicate that the ice sheet over Europe (Hoddell et al., 2008) could have been more extended and not the Laurentide ice sheets in North America. In addition, the model-based reconstruction that we used shows relatively small changes of sea level equivalent between interglacials. Data reconstructions seem to indicate possible larger differences between interglacials (Spratt and Lisiecki, 2016), whose effect on the size of the land surface and the carbon cycle remains to be tested.

## 4 Conclusions

    Using a fully coupled climate model including an interactive carbon cycle, we have shown that the difference between pre-MBE and post-MBE cannot be explained by the simulated changes in ocean and vegetation induced by orbital and greenhouse gases forcing. While the oceanic response alone is in qualitative agreement with data (sign of the changes,

correlation between each interglacial), it largely underestimates the amplitude of the changes. Furthermore, accounting for the vegetation response complicates the simulated response and entirely removes the qualitative agreement. The vegetation response depends on ice sheet extent and accounting for ice sheet variations limits the disagreement.

    Comparison of vegetation changes with available pollen data indicates partial agreement, underlying the need to improve vegetation simulations and increase the data coverage to constrain more precisely the change of vegetation cover. We argue

that additional processes need to be accounted for or should be better represented in climate models to explain the observations. It is either possible that many different processes, some of them not included in the present model, adds up to lead to the observed atmospheric $CO_2$ concentration, or just that a first order process is mis-represented or not included. In



particular, the storage of carbon in frozen soils (permafrost) should be included in future modelling work. Response of the Southern Ocean, the widest oceanic region with large air-sea fluxes of $CO_2$ is also a good candidate, given the known deficiency in coarse resolution climate models for the representation of key element of its dynamics (eddies, katabatic winds, AABW formation, brines…). The use of higher resolution models in this region could help to better evaluate its response to

different interglacial conditions.

**Acknowledgments**

The research leading to these results has received funding from the European Union's Horizon 2020 research and innovation programme under grant agreement No 656625, project 'CHOCOLATE'. We also acknowledge WarmClim, a LEFE-INSU

IMAGO project. All the simulations have been performed on avakas machine from the "Mésocentre de Calcul Intensif Aquitain (MCIA)". We thank Vincent Marieu for his assistance to set up the model on this super computer. Discussions with Anne-Sophie Kremer and Thibaut Caley were useful to mature this paper and are acknowledged.

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

**Captions**

**Figure 1: (a) Atmospheric $CO_2$ (ppm) evolution from data (Luthi et al., 2008), (b) insolation (W/m2) (a) at 65°N on 21[st] of June and (c) at 65°S on 21[st] of December, based on Berger et al. (1978).**

**Figure 2: Ice sheet elevation (m) in the North hemisphere simulated by the CLIMBER-2 model (Ganopolski and Calov, 2011) and used in the OVIC series for each interglacial simulation, in difference with the pre-industrial elevation.**

**Figure 3: Annual SST (°C) in (a) the pre-industrial control simulation and (b-j) the interglacial simulations of the OC series with fixed vegetation and fixed ice sheets, in anomalies with respect to the pre-industrial control simulation.**





**Figure 4: (a) Annual SST difference (°C), (b) Meridional Overturning Circulation difference (Sv) and (c) Dissolved Inorganic Carbon difference (µmol/kg) between pre-MBE (MIS 13, 15, 17, 19) and post-MBE (MIS 5, 7, 9, 11) interglacials simulations for the OC series with fixed vegetation and fixed ice sheets. The vertical black line indicates the limit between the Southern Ocean south of 32°S and the Atlantic Ocean north of 32°S. The dots on panel (a) are SST data differences based on Past Interglacials Working Group of PAGES (2016) (Table 3).**

**Figure 5: Meridional Overturning Circulation (Sv) in the Southern Ocean and in the Atlantic Ocean north of 32°S in (a) the pre-industrial control simulation and, (b-j) the interglacial simulations of the OC series with fixed vegetation and fixed ice sheets, in anomalies with respect to the pre-industrial control simulation. The vertical black line indicates the limit between the Southern Ocean south of 32°S and the Atlantic Ocean north of 32°S.**

**Figure 6: Simulated $CO_2$ in the interglacial simulations of the OC series as a function of the measured $CO_2$ from data (Luthi et al., 2008). The Pearson correlation coefficient and the p-value are indicated on top.**

**Figure 7: Tree cover (%) change with respect to the pre-industrial control simulation for the OVC series with interactive vegetation and fixed ice sheets.**

**Figure 8: Annual sea surface temperature difference (°C) between simulations with interactive vegetation (OVC) and with fixed vegetation (OC).**

**Figure 9: Meridional Overturning Circulation difference (Sv) between simulations with interactive vegetation (OVC) and with fixed vegetation(OC). The vertical black line indicates the limit between the Southern Ocean south of 32°S and the Atlantic Ocean north of 32°S.**

**Figure10: Carbon stocks (GtC) in the three reservoirs (atmosphere, ocean and land) for each simulation. (a) OC series with fixed vegetation and fixed ice sheets, (b) OVC series with interactive vegetation and fixed ice sheets and (c) OVIC series with interactive vegetation and different prescribed ice sheets. The stocks are given as anomalies with respect to the control pre-industrial simulation.**

**Figure 11: (a) $CO_2$ concentration (ppm) at the end of the simulations and (b) composite (average) $CO_2$ (ppm) in the pre-MBE (MIS 13, 15, 17, 19) and post-MBE (MIS 5, 7, 9, 11) interglacial simulations.**

**Figure 12: (a) Tree cover (%) and (b) carbon storage (kgC/m$^2$) difference between pre-MBE (MIS 13, 15, 17, 19) and post-MBE (MIS 5, 7, 9, 11) interglacials simulations for the OVC series with interactive vegetation and fixed ice sheets. Qualitative indication of tree cover change from data are indicated with dots: blue indicates a reduction of tree cover on average during pre-MBE interglacials compared to post-MBE interglacials, and red an increase.**





**Figure 13: (a) Annual sea surface temperature difference (°C), (b) Meridional Overturning Circulation difference (Sv) and (c) Dissolved Inorganic Carbon difference (μmol/kg) between the average of the pre-MBE (MIS 13, 15, 17, 19) and post-MBE (MIS 5, 7, 9, 11) interglacials with interactive vegetation (OVC). The vertical black line indicates the limit between the Southern Ocean south of 32°S and the Atlantic Ocean north of 32°S. The dots on panel (a) are SST data differences based on Past Interglacials Working Group of PAGES (2016) (Table 3).**

**Figure 14: Ice sheet elevation difference (m) between the average of the pre-MBE (MIS 13, 15, 17, 19) and post-MBE (MIS 5, 7, 9, 11) interglacial simulations.**

**Figure 15: Difference of tree cover (%) between the average of the pre-MBE (MIS 13, 15, 17, 19) and post-MBE (MIS 5, 7, 9, 11) interglacials with interactive vegetation and difference ice sheets (OVIC). Qualitative indication of tree cover change from data are indicated with dots: blue indicates a reduction of tree cover on average during pre-MBE interglacials compared to post-MBE interglacials, and red an increase.**

**Figure 16: Annual sea surface temperature difference (°C) between the average of the pre-MBE (MIS 13, 15, 17, 19) and post-MBE (MIS 5, 7, 9, 11) interglacials with interactive vegetation and different ice sheets (OVIC). The vertical black line indicates the limit between the Southern Ocean south of 32°S and the Atlantic Ocean north of 32°S. The dots on panel (a) are SST data differences based on Past Interglacials Working Group of PAGES (2016) (Table 3).**

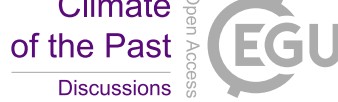



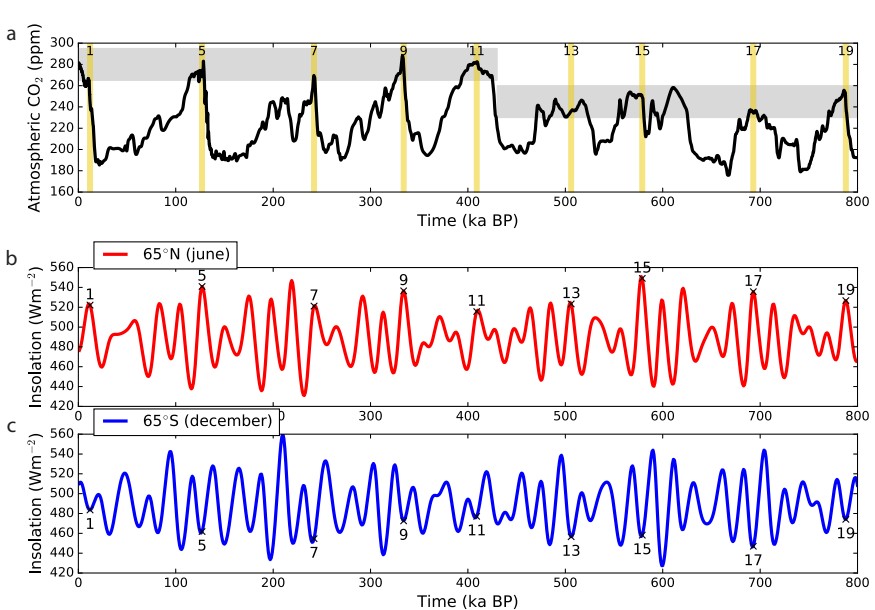

Figure 1: (a) Atmospheric $CO_2$ (ppm) evolution from data (Luthi et al., 2008), (b) insolation (W/m$^2$) (a) at 65 °N on 21st of June and (c) at 65 °S on 21st of December, based on Berger et al. (1978).





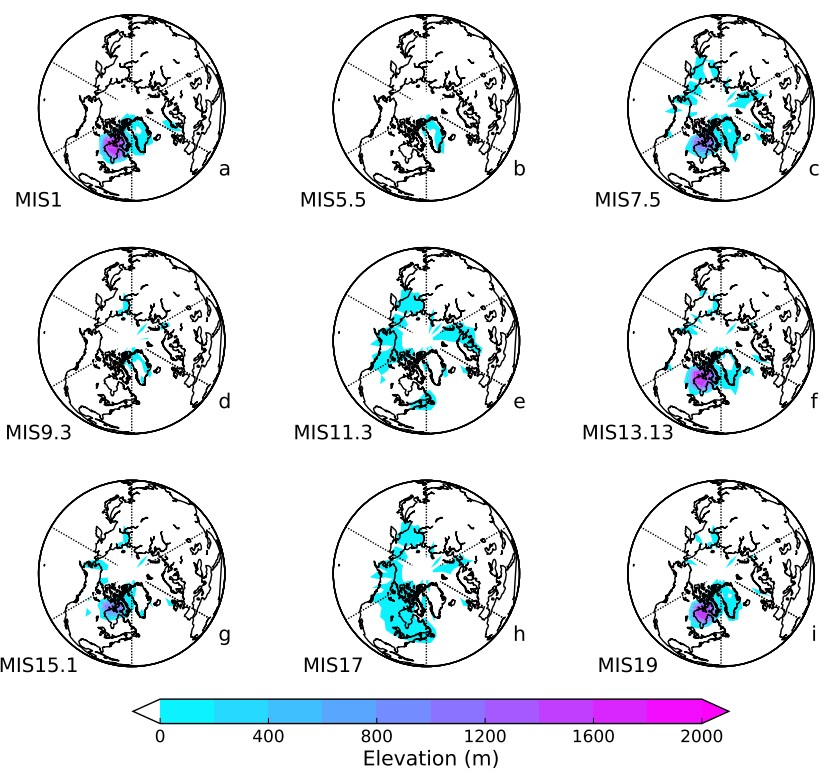

Figure 2: Ice sheet elevation (m) in the North hemisphere simulated by the CLIMBER-2 model (Ganopolski and Calov, 2011) and used in the OVIC series for each interglacial simulation, in difference with the pre-industrial elevation.

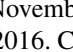


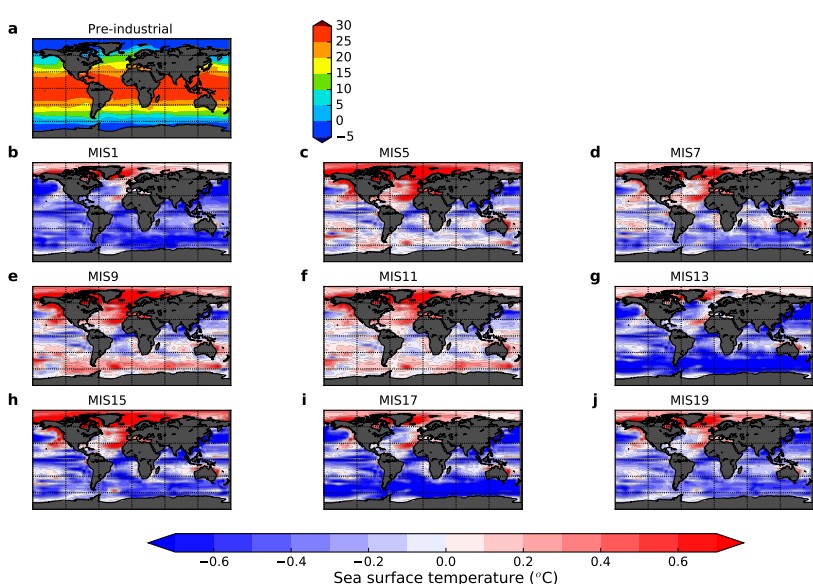

Figure 3: Annual SST (˚C) in (a) the pre-industrial control simulation and (b-j) the interglacial simulations of the OC series with fixed vegetation and fixed ice sheets, in anomalies with respect to the pre-industrial control simulation.





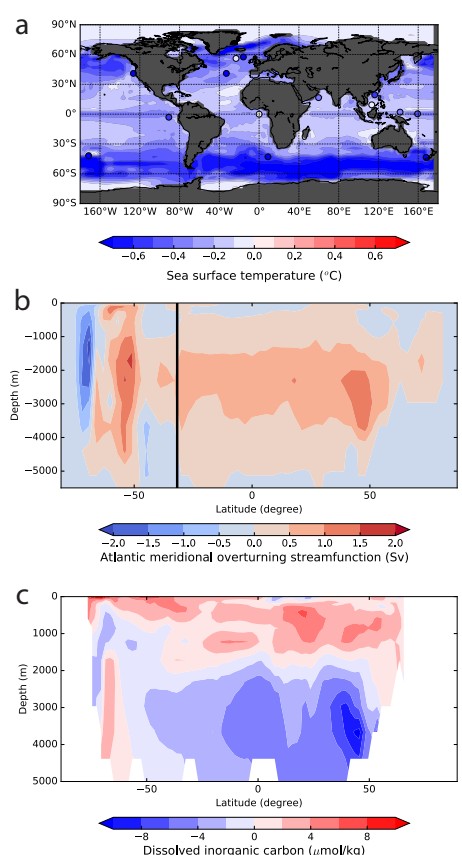

Figure 4: (a) Annual SST difference (˚C), (b) Meridional Overturning Circulation difference (Sv) and (c) Dissolved Inorganic Carbon difference ($\mu$mol/kg) between pre-MBE (MIS 13, 15, 17, 19) and post-MBE (MIS 5, 7, 9, 11) interglacials simulations for the OC series with fixed vegetation and fixed ice sheets. The vertical black line indicates the limit between the Southern Ocean south of 32˚S and the Atlantic Ocean north of 32˚S. The dots on panel (a) are SST data differences based on Past Interglacials Working Group of PAGES (2016) (Table 3).

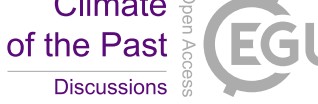



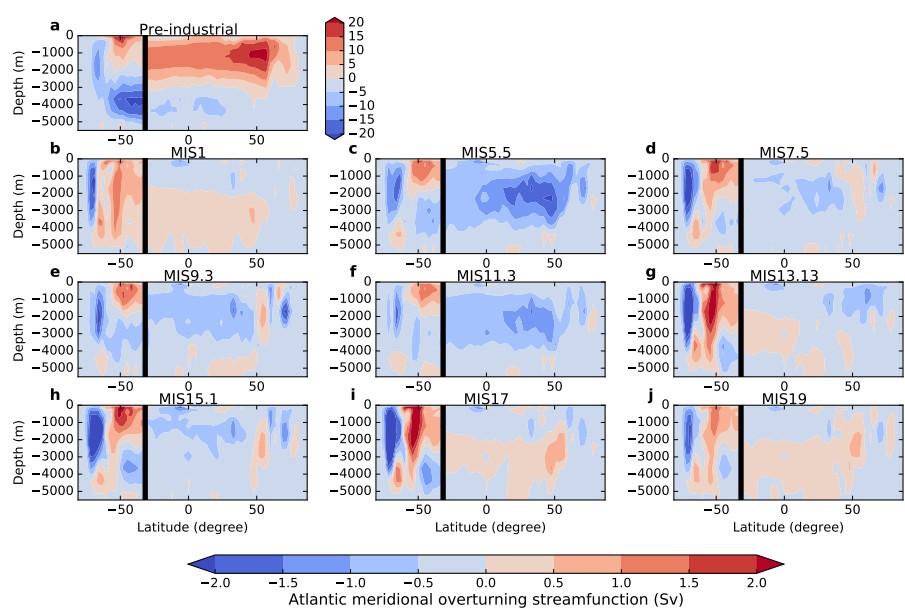

Figure 5: Meridional Overturning Circulation (Sv) in the Southern Ocean and in the Atlantic Ocean north of $32°$ S in (a) the pre-industrial control simulation and, (b-j) the interglacial simulations of the OC series with fixed vegetation and fixed ice sheets, in anomalies with respect to the pre-industrial control simulation. The vertical black line indicates the limit between the Southern Ocean south of $32°$ S and the Atlantic Ocean north of $32°$ S.

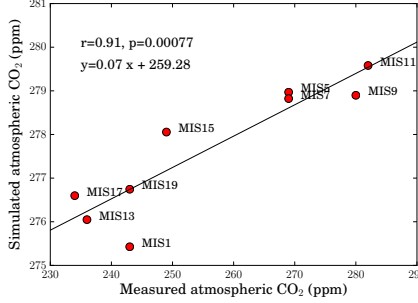

Figure 6: Simulated $CO_2$ in the interglacial simulations of the OC series as a function of the measured $CO_2$ from data (Luthi et al., 2008). The Pearson correlation coefficient and the p-value are indicated on top.





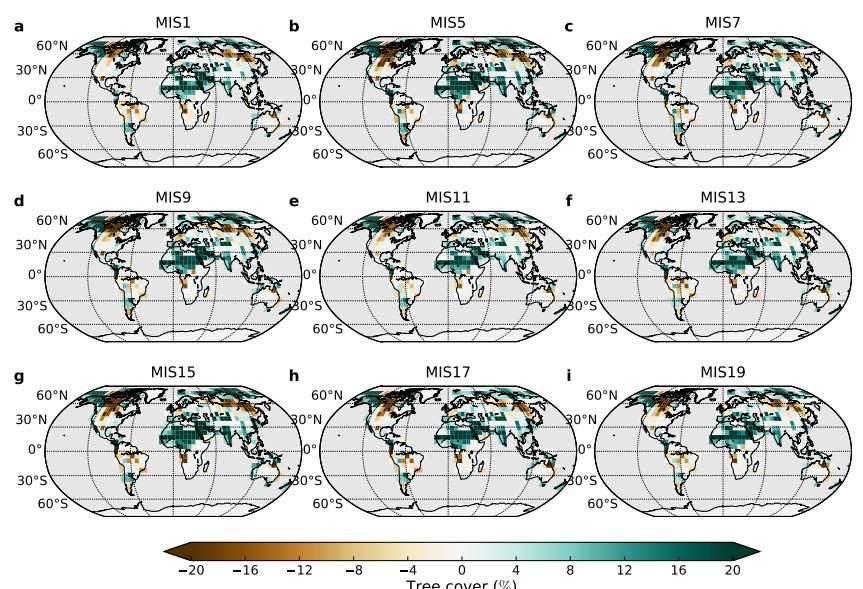

Figure 7: Tree cover (%) change with respect to the pre-industrial control simulation for the OVC series with interactive vegetation and fixed ice sheets.





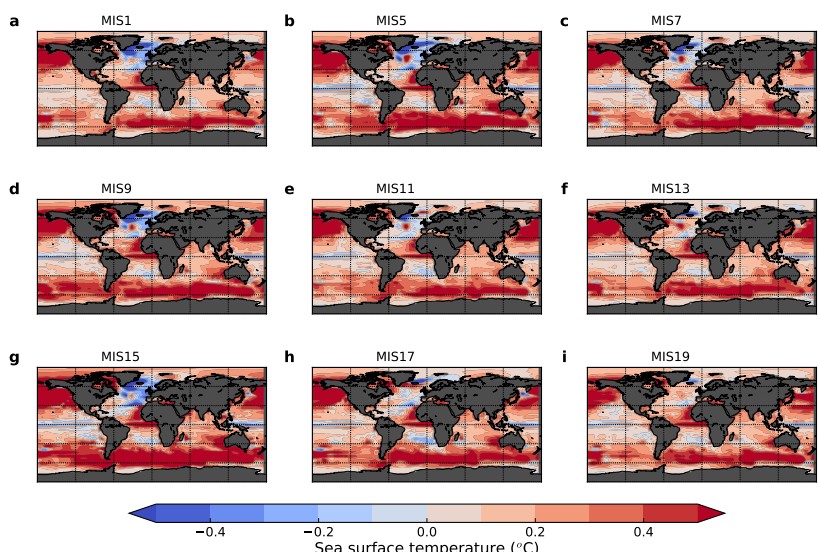

Figure 8: Annual sea surface temperature difference (°C) between simulations with interactive vegetation (OVC) and with fixed vegetation (OC).





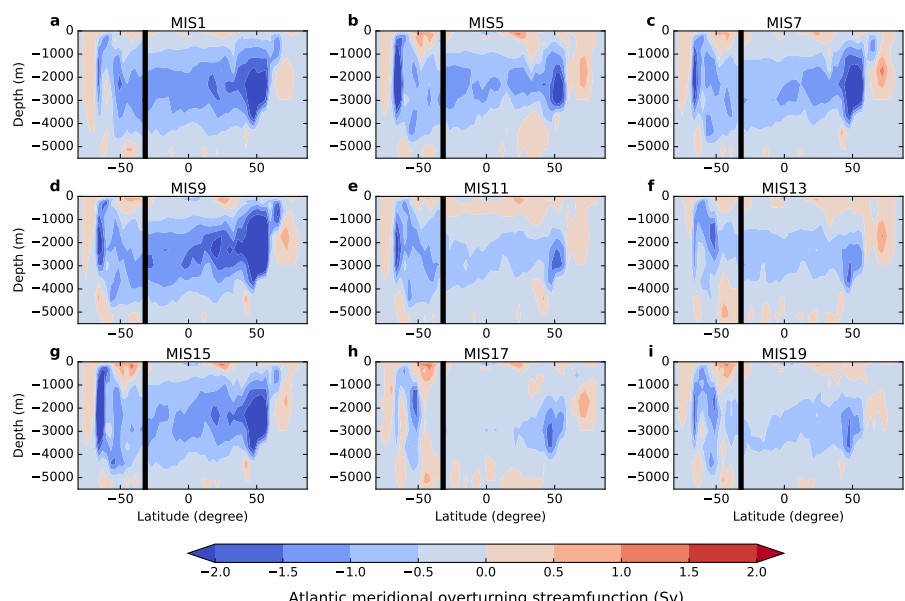

Figure 9: Meridional Overturning Circulation difference (Sv) between simulations with interactive vegetation (OVC) and with fixed vegetation(OC). The vertical black line indicates the limit between the Southern Ocean south of 32 ° S and the Atlantic Ocean north of 32 ° S.





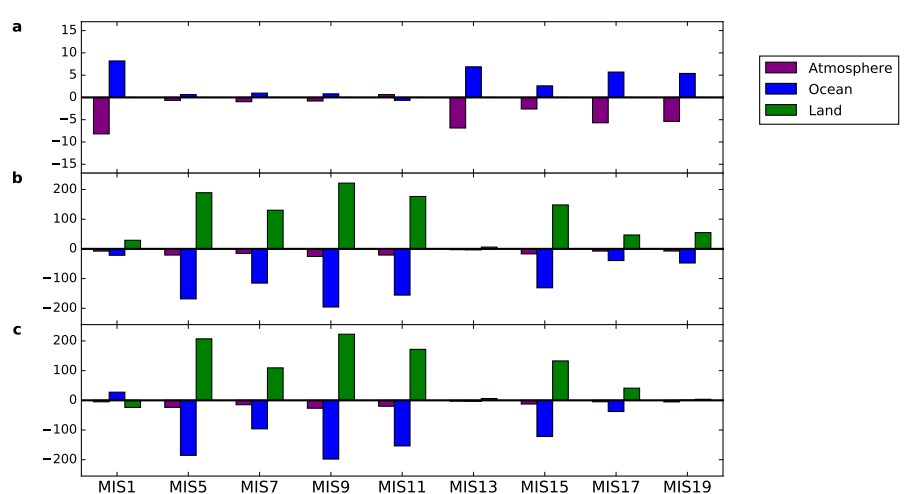

Figure 10: Carbon stocks (GtC) in the three reservoirs (atmosphere, ocean and land) for each simulation. (a) OC series with fixed vegetation and fixed ice sheets, (b) OVC series with interactive vegetation and fixed ice sheets and (c) OVIC series with interactive vegetation and different prescribed ice sheets. The stocks are given as anomalies with respect to the control pre-industrial simulation.



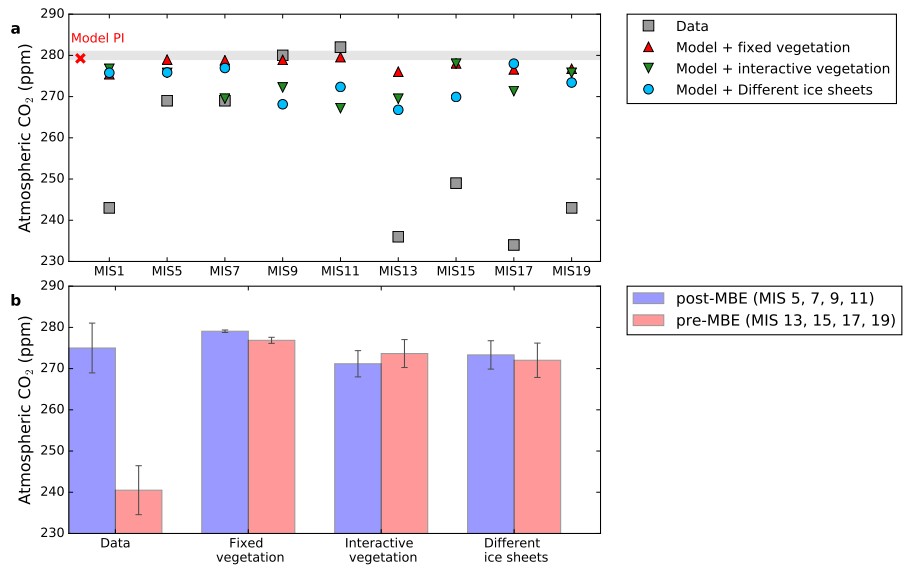

Figure 11: (a) $CO_2$ concentration (ppm) at the end of the simulations and (b) composite (average) $CO_2$ (ppm) in the pre-MBE (MIS 13, 15, 17, 19) and post-MBE (MIS 5, 7, 9, 11) interglacial simulations.





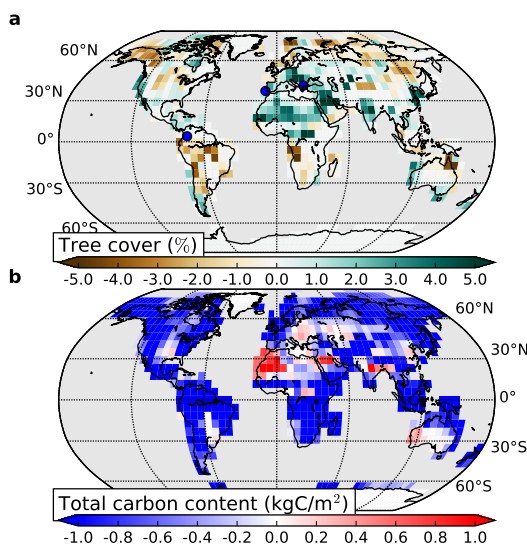

Figure 12: (a) Tree cover (%) and (b) carbon storage (kgC/m$^2$) difference between pre-MBE (MIS 13, 15, 17, 19) and post-MBE (MIS 5, 7, 9, 11) interglacials simulations for the OVC series with interactive vegetation and fixed ice sheets. Qualitative indication of tree cover change from data are indicated with dots: blue indicates a reduction of tree cover on average during pre-MBE interglacials compared to post-MBE interglacials, and red an increase.



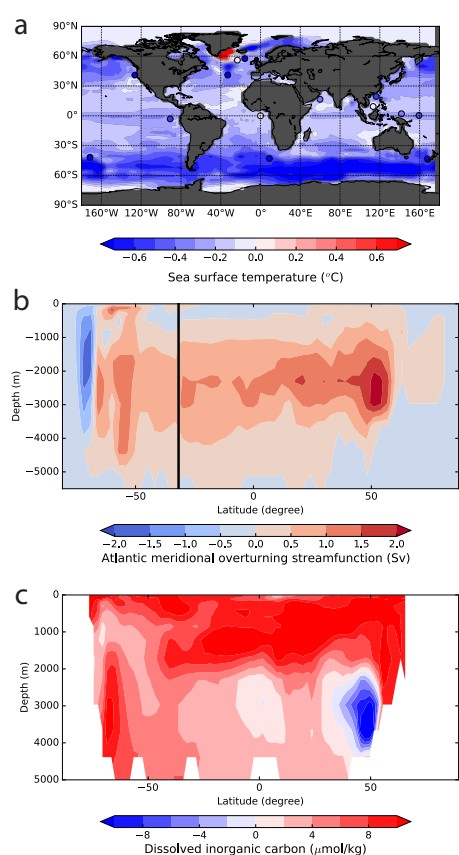

Figure 13: (a) Annual sea surface temperature difference (˚C), (b) Meridional Overturning Circulation difference (Sv) and (c) Dissolved Inorganic Carbon difference ($\frac{1}{4}$mol/kg) between the average of the pre-MBE (MIS 13, 15, 17, 19) and post-MBE (MIS 5, 7, 9, 11) interglacials with interactive vegetation (OVC). The vertical black line indicates the limit between the Southern Ocean south of 32˚S and the Atlantic Ocean north of 32˚S. The dots on panel (a) are SST data differences based on Past Interglacials Working Group of PAGES (2016) (Table 3).





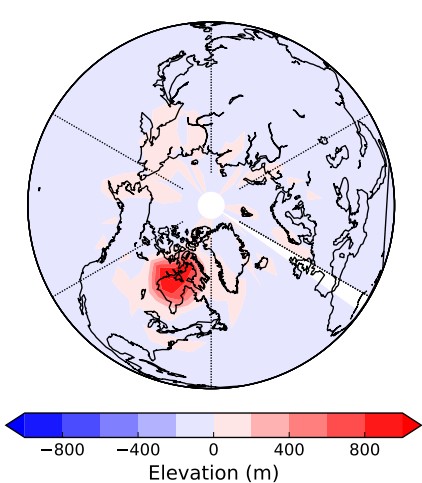

Figure 14: Ice sheet elevation difference (m) between the average of the pre-MBE (MIS 13, 15, 17, 19) and post-MBE (MIS 5, 7, 9, 11) interglacial simulations.





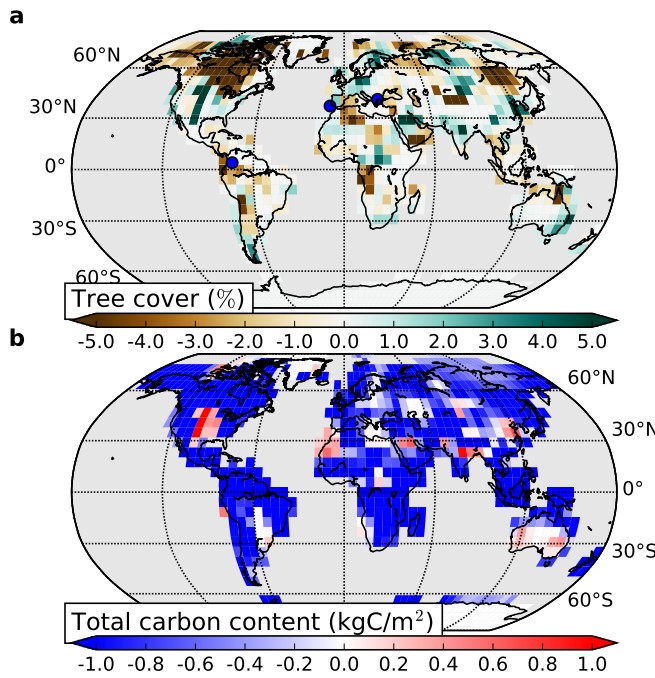

Figure 15: Difference of tree cover (%) between the average of the pre-MBE (MIS 13, 15, 17, 19) and post-MBE (MIS 5, 7, 9, 11) interglacials with interactive vegetation and difference ice sheets (OVIC). Qualitative indication of tree cover change from data are indicated with dots: blue indicates a reduction of tree cover on average during pre-MBE interglacials compared to post-MBE interglacials, and red an increase.



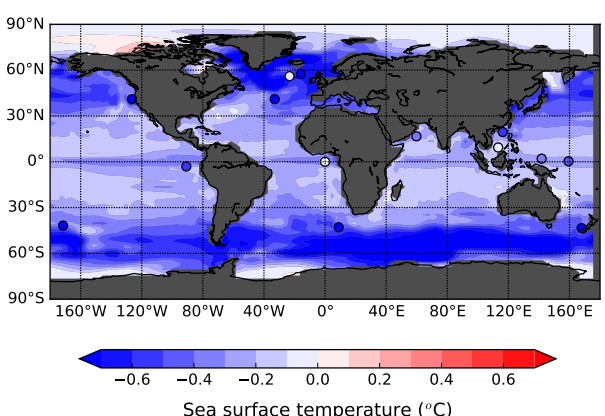

Figure 16: Annual sea surface temperature difference (°C) between the average of the pre-MBE (MIS 13, 15, 17, 19) and post-MBE (MIS 5, 7, 9, 11) interglacials with interactive vegetation and different ice sheets (OVIC). The vertical black line indicates the limit between the Southern Ocean south of 32°S and the Atlantic Ocean north of 32°S. The dots on panel (a) are SST data differences based on Past Interglacials Working Group of PAGES (2016) (Table 3).