# Peer review of "Response of the carbon cycle to the different orbital configurations of the last 9 interglacials"

_Climate of the Past, 2016_

## Referee Comment (RC1) · Anonymous Referee #1 · 12 Dec 2016

This is a well-conceived study about an interesting and relevant topic. The methodology is sound, and the fact that the authors' model could not reproduce the observed changes in CO2 before and after the Mid-Bruhnes event (MBE) should not prevent it from being published.

However, this manuscript needs a background section describing in more detail the previous studies that have addressed this question and the hypotheses that have been proposed (e.g., by Yin and Berger and Kohler). At the end of the manuscript the authors should revisit these hypotheses. Do the new model results presented support either hypothesis (ie, that stronger or weaker overturning explains the change in CO2)? More background information about the ice sheet model used would also be helpful. What sea level is simulated for each interglacial?

[Figure]

In the results section, it would be useful to have a more specific comparison of the proxy and model SST changes. The authors have a very nice table summarizing proxy SST observations, but it isn't clear how well the model agrees with the data. I can't tell in the figures how large the model SST changes are. How much beyond -0.6 C does the dark blue color go? Simply listing the global mean SST change as well as values for the North Atlantic and Southern Ocean would be helpful.

In their discussion, the authors suggest that the reason that the model did not reproduce large enough $CO_2$ changes could be related to a shortcoming in how it simulates bottom water formation. Additionally, the authors identify mismatches between proxy and simulated vegetation changes. They should provide more information related to these potential problems. How well does the model simulate the Holocene or pre-industrial with respect to atm $CO_2$ level, overturning and vegetation? Can the authors suggest more specific solutions to address these shortcomings? Are there additional simulations, such as sensitivity tests, that the authors could propose (or run) to gain more insights?

Lastly, I think the manuscript has too many figures. Several figures could be combined to make it easier to compare the different simulation scenarios. For example, Figure 4 could have 3 columns, one each for the OC, OVC, and OVIC simulations (thus, combining figures 4, 13, and 16). Similarly, results from figures 12 and 15 could be placed side-by-side.

---

## Referee Comment (RC2) · Anonymous Referee #2 · 22 Feb 2017

Bouttes et al did an excellent job running a coupled EMIC model with carbon cycles to study the model's response to the different climate conditions of the last 9 interglacials. Unfortunately, even though the data show about 35ppm changes among the 9 interglacials, the model can only produce about 4 ppm changes, and the authors conclude that the fail of reproducing the 35 ppm is due to "mis-representation of some key processes in the model".

First, I suggest that the title needs to be changed to something like below to better represent the major topic of this paper.

"Response of the carbon cycle in an intermediate complexity model to the different climate configuration of the last 9 interglacials".

So the readers know that it's a model's carbon cycle response and doesn't imply that

the response is derived from the data. Also since this study did simulations with the different orbital, vegetation and ice sheets, so it's better to use climate configuration other than orbital configuration in the title.

Second, the authors need to explain how the increase of vegetation on land can produce HUGE global ocean warming (Figure 8).

Third, MIS17 in Figure 2 looks more like glacial instead of interglacial.

Fourth, P10L22, Change "Using a fully coupled climate model" to "Using a fully coupled climate model with an intermediate complexity"

---

## Author Comment (AC1) · 27 Oct 2017

**Response to reviewer #1**

We thank the reviewer for their comments. In the following we respond point by point.

**Anonymous Referee #1**
This is a well-conceived study about an interesting and relevant topic. The methodology is sound, and the fact that the authors' model could not reproduce the observed changes in CO2 before and after the Mid-Bruhnes event (MBE) should not prevent it from being published.

1. However, this manuscript needs a background section describing in more detail the previous studies that have addressed this question and the hypotheses that have been proposed (e.g., by Yin and Berger and Kohler).
Very few modelling studies have looked at the difference of atmospheric $CO_2$ before and after the MBE. Yin and Berger (2010, 2012) and Yin (2013) focused on the change of climate and ocean circulation, and Yin (2013) suggested that the change of ventilation could play a role in different $CO_2$ levels, but this remained to be tested. Kohler and Fischer looked at the different interglacial $CO_2$ values, but with a simple box model. As suggested we have expanded this section on previous work in the introduction with more details:
"To explain the different climates of interglacials before and after the MBE, modelling studies have shown that it is necessary to include the change of atmospheric $CO_2$ (Yin and Berger, 2010; 2012). Indeed, these numerical simulations with an intermediate complexity model have demonstrated that differences in Earth's orbital configuration, and hence seasonal and spatial distribution of insolation, cannot explain alone the colder climate recorded during pre-MBE interglacials, whereby lower atmospheric $CO_2$ concentration is also necessary to simulate colder climate (Yin and Berger, 2010; 2012). However, the reasons for the lower $CO_2$ values remain elusive and very few modelling exercises have tackled the issue of different $CO_2$ levels during interglacials before and after the MBE. Köhler and Fischer (2006) have produced transient simulations of the last 740,000 years using the BICYCLE box model. They used several paleoclimatic records such as ocean temperature, sea ice, sea level, ocean circulation, marine biota, terrestrial biosphere and $CaCO_3$ chemistry to force forward their model. They run a set of simulations prescribing only one forcing at a time and another with all forcings excluding one at a time, which allows them to analyse which forcings are the most important. In their simulations, they have shown that the lower $CO_2$ values during pre-MBE are mainly explained by the prescribed lower Southern Ocean (SO) sea surface temperature and weaker Atlantic meridional overturning circulation (low North Atlantic Deep Water formation and SO vertical mixing) compared to post-MBE interglacials. Using an intermediate complexity model, Yin (2013) conversely simulated vigorous bottom water formation and stronger ventilation in the Southern Ocean during pre-MBE interglacials and suggested this could increase deep oceanic carbon storage and lower atmospheric $CO_2$. However, this effect on the ocean carbon reservoir and atmospheric $CO_2$ has not been evaluated yet in a climate model including a carbon cycle representation."

2. At the end of the manuscript the authors should revisit these hypotheses. Do the new model results presented support either hypothesis (ie, that stronger or weaker overturning explains the change in CO2)?

In this work, we show that the modelled increase in overturning in response to the different interglacial orbital forcings and $CO_2$ is too small to yield much effect on the ocean carbon storage, and results in very small changes in atmospheric $CO_2$, with appropriate tendency as compared to observations, but an order of magnitude too small (Fig. 11). Because the ocean circulation change is small, it does not allow to decipher between the two hypotheses, which could also both be wrong, but shows that either the change of ventilation simulated by the model is not correct, or that other processes impacting the carbon cycle are missing.

In the first hypothesis, the change of $CO_2$ could still be due to changes in circulation, and whether it is due to stronger or weaker overturning could be tested with sensitivity experiments. Yet this would ultimately require a mechanism yielding such changes of circulation. Alternatively, the lower atmospheric $CO_2$ before the MBE could be due to other processes on land or in the ocean. To present these different ideas, we have added in the conclusion of the manuscript:

"Past work suggested that either a vigorous AABW (Yin, 2013) or weak Atlantic thermohaline circulation (Köhler and Fischer, 2006) during pre-MBE interglacials could increase the oceanic carbon storage and explain the lower $CO_2$ than during post-MBE interglacials. Other studies for different background climates have shown opposite results with respect to the effect of ocean circulation on carbon storage. A weaker AMOC could either result in more ocean carbon storage with a pre-industrial climate (Obata, 2007; Menviel et al., 2008; Bozbiyik et al., 2011) or glacial climate (Menviel et al., 2008), or it could yield less ocean carbon storage with a pre-industrial climate (Marchal et al., 1998; Swingedouw et al., 2007; Bouttes et al., 2012) or a glacial climate (Schmittner and Galbraith, 2008; Bouttes et al., 2012; Schmittner and Lund, 2015;). Menviel (2014) showed that on top of changes in NADW formation, modifications of AABW and NPDW formations could results in different oceanic carbon storage. Data indicate that the modern reduction of carbon uptake in the North Atlantic is due to a reduction in the overturning circulation (Perez et al., 2013). Because the atmospheric $CO_2$ change that we simulate has a low magnitude of only a few ppm, it is not yet possible to infer whether stronger or weaker overturning during pre-MBE interglacials could have significantly lowered atmospheric $CO_2$."

3. More background information about the ice sheet model used would also be helpful. What sea level is simulated for each interglacial?

In this study, we didn't use an ice sheet model but only the outputs from another coupled climate-ice sheet model (Ganopolski and Calov, 2011) to prescribe the ice sheets, as no reconstructed ice sheet from data exist for the nine last interglacials. We have added more details on this model:

"The prescribed ice sheet distributions are thus taken from an ice sheet simulation of the last 800,000 years (Ganopolski and Calov, 2011) using the intermediate complexity model CLIMBER-2 (Petoukhov et al., 2000; Ganopolski et al., 2001; Brovkin et al., 2002), including a 3-D polythermal ice sheet model (Greve, 1997). This ice sheet model is coupled to the climate component via surface energy and mass balance interface (Calov et al., 2005), which accounts for the effect of aeolian dust deposition on snow albedo."

We have also added the corresponding sea levels for each interglacial at the dates chosen for the snapshot experiments in Table 1.

| MIS | Date of $\delta^{18}O$ peak (ka BP) | Date for orbital configuration and $CO_2$ (ka BP) | $CO_2$ values from data (ppm) | Sea level changes (m) corresponding to the ice sheet configurations |
|---|---|---|---|---|
| 1 | 6 | 12 | 243.2 | 13.8 |
| 5.5 | 123 | 127 | 268.64 | -0.8 |
| 7.5 | 239 | 242 | 269.23 | 5.6 |
| 9.3 | 329 | 334 | 280.32 | -0.9 |
| 11.3 | 405 | 409 | 282.29 | -0.8 |
| 13.13 | 501 | 506 | 235.92 | 13.1 |
| 15.1 | 575 | 579 | 249.36 | 2.3 |
| 17 | 696 | 693 | 234.38 | -0.4 |
| 19 | 780 | 788 | 242.73 | 10.8 |

Table 1 Dates of orbital parameters and $CO_2$ used for the simulations (Luthi et al., 2008), and sea level anomalies as compared to present-day conditions (m) corresponding to the prescribed ice sheets (Ganopolski and Calov, 2011).

4. In the results section, it would be useful to have a more specific comparison of the proxy and model SST changes. The authors have a very nice table summarizing proxy SST observations, but it isn't clear how well the model agrees with the data. I can't tell in the figures how large the model SST changes are. How much beyond -0.6 C does the dark blue color go? Simply listing the global mean SST change as well as values for the North Atlantic and Southern Ocean would be helpful.
The dark blue color is for all values below -0.6°C. As suggested we have listed the global mean SST change and the values for the North Atlantic and Southern Ocean in the text. The values are summarized below (in °C):

| | OC | OVC | OVIC |
|---|---|---|---|
| Global | -0.30 | -0.28 | -0.32 |
| North Atlantic (30°N-65°N) | -0.36 | -0.31 | -0.49 |
| Southern Ocean (south of 54°S) | -0.43 | -0.44 | -0.47 |

5. In their discussion, the authors suggest that the reason that the model did not reproduce large enough CO2 changes could be related to a shortcoming in how it simulates bottom water formation. Additionally, the authors identify mismatches between proxy and simulated vegetation changes. They should provide more information related to these potential problems. How well does the model simulate the Holocene or preindustrial with respect to atm CO2 level, overturning and vegetation? Can the authors suggest more specific solutions to address these shortcomings? Are there additional simulations, such as sensitivity tests, that the authors could propose (or run) to gain more insights?

The carbon cycle module in iLOVECLIM has been validated for the pre-industrial (Bouttes et al., 2015) but not tested for the Holocene. The overturning and vegetation have been described and validated by Goose et al. (2010)

The terrestrial biosphere module in iLOVECLIM, Vecode, is very simple with only two plant functional types. To test the impact of different vegetation responses to orbital forcings and $CO_2$ from the different interglacials, the vegetation distribution could be obtained from a more complex model and then prescribed in iLOVECLIM.

Concerning the overturning, it could be artificially modified by adding fresh water or using a scheme for the sinking of brines from sea ice as in Bouttes et al. (2009).

Finally, another test concerns the ice sheets, which are not well constrained at all for these periods of time. Sensitivity experiments could be run with prescribed ice sheets designed to be very different and idealized to evaluate their impact.

We have added a discussion on these potential additional sensitivity tests, which remain far too extensive for this already long paper (lots of figures as already noticed by the reviewers), but should constitute an interesting follow up to be done later on.

In part 3.3: "In addition, the model-based reconstruction that we used shows relatively small changes of sea level equivalent between interglacials. Data reconstructions seem to indicate possible larger differences between interglacials (Spratt and Lisiecki, 2016), whose effect on the size of the land surface and the carbon cycle remains to be tested. Sensitivity experiments with prescribed idealised ice sheets designed to be very different would help to evaluate their impact."

In the conclusion:

"The vegetation model in iLOVECLIM only simulates grass and trees, to better evaluate the different vegetation response to orbital and $CO_2$ forcings it would be useful to use a more complex terrestrial biosphere model. "

"The impact of ventilation changes could be tested by artificially modifying the buoyancy forcing in the areas of bottom water formation."

Bouttes, N., D. Paillard and D. M. Roche, Impact of brine-induced stratification on the glacial carbon cycle, Clim. Past, 6, 575-589, doi: 10.5194/cp-6-575-2010, 2010

6. Lastly, I think the manuscript has too many figures. Several figures could be combined to make it easier to compare the different simulation scenarios. For example, Figure 4 could have 3 columns, one each for the OC, OVC, and OVIC simulations (thus, combining figures 4, 13, and 16). Similarly, results from figures 12 and 15 could be placed side-by-side.

As suggested we have combined figures 4 and 13 together but we have left figure 16 alone as it has only one panel and it would have made the space taken by figures larger.

OC: fixed vegetation, fixed ice sheets          OVC: interactive vegetation, fixed ice sheets

[Figure]

We have also combined figures 12 and 15 together.

[Figure]

**References**

Bouttes, N., Roche, D. M., and Paillard, D.: Systematic study of the impact of fresh water fluxes on the glacial carbon cycle, Clim. Past, 8, 589-607, 2012.

Bozbiyik, A., Steinacher, M., Joos, F., Stocker, T. F., and Menviel, L.: Fingerprints of changes in the terrestrial carbon cycle in response to large reorganizations in ocean circulation, Clim. Past, 7, 319–338, doi:10.5194/cp-7-319-2011, 2011.

Brovkin, V., Bendtsen, J., Claussen, M., Ganopolski, A., Kubatzki, C., Petoukhov, V., and Andreev, A.: Carbon cycle, vegetation and climate dynamics in the Holocene: experiments with the CLIMBER-2 model, Global Biogeochem. Cyc., 16, 1139, doi:10.1029/2001GB001662, 2002.

Calov, R., Ganopolski, A., Claussen, M., Petoukhov, V., and Greve. R.: Transient simulation of the last glacial inception, Part I: Glacial inception as a bifurcation of the climate system, Clim. Dynam., 24, 545–561, doi:10.1007/s00382-005-0007-6, 2005.

Ganopolski, A., Petoukhov, V., Rahmstorf, S., Brovkin, V., Claussen, M., Eliseev, A., and Kubatzki, C.: CLIMBER-2: a

climate system model of intermediate complexity, Part II: Model sensitivity, Clim. Dynam., 17, 735–751, 2001.

Greve, R.: A continuum-mechanical formulation for shallow poly- thermal ice sheets, Philos. T. Roy. Soc. Lond. A, 355, 921–974, 1997.

Marchal, O., Stocker, T. F., and Joos, F.: Impact of oceanic reor- ganizations on the ocean carbon cycle and atmospheric carbon dioxide content, Paleoceanography, 13, 225–244, 1998.

Menviel, L., Timmermann, A., Mouchet, A., and Timm, O.: Meridional reorganizations of marine and terrestrial productivity during Heinrich events, Paleoceanography, 23, PA1203, doi:10.1029/2007PA001445, 2008.

Menviel, L., England, M.H., Meissner, K., Mouchet, A., Yu, J.: Atlantic-Pacific seesaw and its role in outgassing $CO_2$ during Heinrich events, Paleoceanography, 29, doi:10.1002/2013PA002542, 2014.

Obata, A.: Climate-Carbon Cycle Model Response to Freshwater Discharge into the North Atlantic, J. Climate, 20, 5962–5976, doi:10.1175/2007JCLI1808.1, 2007.

Pérez, F. F., Mercier, H., Vázquez-Rodríguez, M., Lherminier, P., Velo, A., Pardo, P. C., Rosón, G. and Ríos, A. F.: Atlantic Ocean $CO_2$ uptake reduced by weakening of the meridional overturning circulation, Nature Geoscience, 6, 146–152, doi: 10.1038/NGEO1680, 2013.

Petoukhov, V., Ganopolski, A., Brovkin, V., Claussen, M., Eliseev, A., Kubatzki, C., and Rahmstorf, S.: CLIMBER-2: A climate system model of intermediate complexity, Part I: Model description and performance for present climate, Clim. Dynam., 16, 1– 17, 2000.

Schmittner, A. and Galbraith, E. D.: Glacial greenhouse-gas fluctuations controlled by ocean circulation changes, Nature, 456, 373–376, doi:10.1038/nature07531, 2008.

Schmittner, A., and Lund, D. C.: Early deglacial Atlantic overturning decline and its role in atmospheric $CO_2$ rise inferred from carbon isotopes ($\delta^{13}$C) Climate of the Past, 11, 135-152, 2015.

Swingedouw D., Bopp L., Matras A. and Braconnot P.: Effect of land-ice melting and associated changes in the AMOC result in little overall impact on oceanic $CO_2$ uptake, Geophysical Research Letters 34, L23706, 2007b.

---

## Author Comment (AC2) · 27 Oct 2017

**Response to reviewer #2**

We are thankful to the reviewer for their comments and we respond point by point in the following.

**Anonymous Referee #2**

Bouttes et al did an excellent job running a coupled EMIC model with carbon cycles to study the model's response to the different climate conditions of the last 9 interglacials. Unfortunately, even though the data show about 35ppm changes among the 9 interglacials, the model can only produce about 4 ppm changes, and the authors conclude that the fail of reproducing the 35 ppm is due to "mis-representation of some key processes in the model".

1. First, I suggest that the title needs to be changed to something like below to better represent the major topic of this paper.
"Response of the carbon cycle in an intermediate complexity model to the different climate configuration of the last 9 interglacials".
So the readers know that it's a model's carbon cycle response and doesn't imply that the response is derived from the data. Also since this study did simulations with the different orbital, vegetation and ice sheets, so it's better to use climate configuration other than orbital configuration in the title.
We agree and have modified the title to the suggested title: "Response of the carbon cycle in an intermediate complexity model to the different climate configurations of the last 9 interglacials"

2. Second, the authors need to explain how the increase of vegetation on land can produce HUGE global ocean warming (Figure 8).
As seen on Figure 8, including an interactive vegetation model leads to warming of the SSTs in most of the ocean. This warming is of a few tenths of a degree, which is relatively modest compared to the glacial-interglacial change of temperature of a few degrees. The change of vegetation modifies the local albedo and evapo-transpiration, in particular in the high NH latitudes where more tree cover leads to reduced albedo and larger transpiration, hence warming. Yet a more detailed analysis with sensitivity experiments taking into account the effect of changing vegetation on only one variables at a time (albedo, evaporation…) and at one region at a time, as well as possible retroactions, would be necessary to pinpoint the exact reasons of the warming, which is beyond the scope of this study.

3. Third, MIS17 in Figure 2 looks more like glacial instead of interglacial.
The figure below shows the ice sheet elevation for MIS17 and the preceding glacial maxima from the model simulations as comparison. While at 693 kaBP the ice sheet covers part of North America, its elevation is very low (a few tens of meters) compared to a glacial period such as at 722 kaBP when most of the ice sheet is higher than 1000 meters.

[Figure]

| MIS17 (693 ka) | 722 ka |

Elevation (m)

Even though MIS17 is different from a glacial, it highlights the need to better constrain the ice sheets. While more data and ice sheet simulations can help, it would also be useful to run sensitivity experiments with different prescribed ice sheet configurations in the carbon-climate model to evaluate the impact of those different ice sheets.

4. Fourth, P10L22, Change "Using a fully coupled climate model" to "Using a fully coupled climate model with an intermediate complexity"
We have modified to: "Using a fully coupled climate model of intermediate complexity".

Additional modification:
In addition, due to recent measurements, we have modified pollen values in table 4 for MIS17 for the Iberian margin, which increases the tree cover there and gives better agreement between model and data (Figure 12), and for MIS15 which do not modify the qualitative results.